# The cost of doing nothing about a sleeper weed–*Nassella neesiana* in New Zealand

Graeme W. Bourdôt[1]*, Christopher E. Buddenhagen[2]

1 AgResearch Limited, Tuhiraki, Lincoln, New Zealand, 2 AgResearch Limited, Ruakura Research Centre, Hamilton, New Zealand

☯ These authors contributed equally to this work.
* graeme.bourdot@agresearch.co.nz

## Abstract

*Nassella neesiana* (Chilean needle grass), an invasive 'sleeper weed' established in sheep and beef pastures in three of New Zealand's sixteen local government regions, has a potential geographic range amounting to 3.96 million hectares spanning all regions except the West Coast. It impacts the productivity, market value and welfare of livestock through its sharp penetrating that cause blindness and the downgrading of wool, hides, and carcasses. In this study we estimate the benefit of preventing its spread as the present value (PV) of local (regional) and national productivity losses that would accrue over 200 years under a 'do nothing' spread scenario. Using a 3% discount rate and two assumed spread rates, 201 and 100 years to 90% occupation of its potential range, we calculate national PV losses of NZ$ 192 million and NZ$ 1,160 million respectively. In a breakeven analysis, these losses, which equate to the national benefits of preventing the spread, justify annual expenditures of NZ$ 5.3 million and NZ$ 34 million respectively. Restricting the analyses to the regions with known infestations (Hawke's Bay, Marlborough, Canterbury) provided much lower estimates of the benefits (ranging from NZ$ 16.8 million to NZ$ 158 million) because spillover benefits from preventing spread to the other susceptible regions are not accounted for. These analyses support a nationally coordinated approach to managing *N. neesiana* in New Zealand involving surveillance and control measures respectively in the susceptible and infested regions.

## Introduction

Successful containment or eradication of an incipient invasive species is usually predicated on its early detection–while it still has a limited distribution [1–3]. For problematic fast spreading invaders, detection rarely occurs while eradication or containment is feasible, but for slower spreading species, eradication or containment may be a realistic prospect many years after naturalisation has occurred. Regardless, the decision to target a species for eradication or containment is rarely taken lightly, and managers need to be confident about the harm it may cause. They also need to be willing to commit sufficient resources to achieve management goals [1] which may not be realised for years or decades into the future e.g., if the species has a long-

**Data Availability Statement:** All relevant data are within the paper. The references below provide a full description of the method and an online application that will enable a reader to fully duplicate our analysis. The input and output data

  

for analysis are in the paper. 23. Bourdôt GW, Basse B, Kriticos DJ, Dodd M. A cost-benefit analysis blueprint for regional weed management: Nassella neesiana (Chilean needle grass) as a case study. New Zealand Journal of Agricultural Research. 2015;58(3):1-14. doi: 10.1080/00288233.2015.1037460. 24. Bourdot G, McAuliffe R. Cost Benefit Analysis for Regional Pest Management 2017. Available from: https://www.agresearch.co.nz/cba/.

**Funding:** GWB, CEB - Supported by the AgResearch 'Internal Biosecurity Against Weeds' project funded by the Strategic Science Investment Fund administered by the New Zealand Ministry of Business, Innovation and Employment. https://www.mbie.govt.nz/science-and-technology/science-and-innovation/funding-information-and-opportunities/investment-funds/strategic-science-investment-fund/ssif-funded-programmes/agresearch/ The funders had no role in study design, data collection and analysis, decision to publish, or preparation of the manuscript.

**Competing interests:** The authors have declared that no competing interests exist.

lived seed bank [4]. Questions are usually raised about the feasibility of achieving eradication and containment goals with realistically available funds. Apart from raising adequate funding, a key to success is the ability of an agency to enforce cooperation from stakeholders [1,5]. They will also rely on identifiable aspects of the basic natural history of the target species that make it vulnerable to the available control options [1]. Ideally stakeholders are fully invested in managing the species as well. Project leaders must be energetic, and persistent in the face of setbacks [1,5].

With eradication or containment of sleeper weeds early in the invasion process, we contend that the point of the work is less about the reduction of an invader's present impacts and more about the prevention of future impacts. Here, using the example of *Nassella neesiana* in New Zealand, we present an economic approach to determining the potential 'cost' of the invasion and the investment in management that would be justified to avoid this cost. While the main benefits of managing sleeper weeds are in preventing their (still unrealized) future impacts, some economists have instead used optimal control theory to identify a tolerable invasion density, or impact level, commensurate with the costs of the control measures [6,7]. By contrast, bioeconomic analysis of invasions reveals that prevention (of spread) is the policy with the greatest long-term net benefit [8]. Here we account explicitly for the effect of the spread expected to occur in the absence of management, a neglected aspect in many analyses of the economics of invasive species management [9,10].

*Nassella neesiana* (Trin. & Rupr.) Barkworth, [synonym *Stipa neesiana*] or Chilean needle grass (family *Poaceae*; sub-family *Pooideae*; tribe *Stipeae*) is a tufted perennial grass native to temperate South America. The species has naturalised in temperate grasslands in New Zealand (Fig 1) where it reduces biodiversity and stock carrying capacity due to the annual production of masses of unpalatable flower stalks [11,12]. Its seeds can cause the downgrading of wool, skins, hides and carcasses due to the sharp callus and hygroscopic geniculate awn that together facilitate penetration into the wool, skin, and underlying muscle of grazing animals [12–14]. Soil seed banks may be large (18,000 seeds m$^{-2}$) and seed decay rates may vary from 38% to 77% loss year$^{-1}$ while long-lived seeds buried at depths greater than 50 mm may survive more than 10 years [15]. The species has to date occupied a very small fraction of its potential geographic range (0.24%) [16] and can therefore be considered a sleeper weed; one that is naturalised, has not yet spread extensively, but maintains the potential to do so [17]. The area of climatically suitable land in New Zealand has been estimated at between 10 and 15 million ha [16,18], and this potential range is predicted to increase by 60% by 2080 due to climate change [18].

*N. neesiana* is the subject of a variety of management initiatives in Australia and New Zealand. In Australia it is declared a "Weed of National Significance" [19] and its sale and distribution is prohibited under the Quarantine Act 1908. Additionally, in the Australian Capital Territory and parts of New South Wales it is a 'Declared Pest Plant' requiring its control by landholders [11]. In New Zealand it is noted in 15 of the 16 Regional Pest Management Plans but is known from only in the three regions, Hawke's Bay, Marlborough, and Canterbury (Fig 1) where it is classified as a 'Sustained Control' plant. This classification requires landholders to control their infestations to prevent escalating impacts and spread in the region. Such a 'containment' approach is appropriate because most long-distance dispersal of *N. neesiana* is via human activities such as the movement of seed-contaminated livestock, hay and farm machinery [12,19,20]. Wild animals and flood waters may contribute to its spread [14] but the seeds are poorly adapted for natural dispersal by wind [11]. The importance of human-mediated dispersal is illustrated by its occurrence in Genoa, Italy, where it is found growing near tanning works that have processed hides from Argentina [21]. In North Canterbury, in the South Island of New Zealand, it was first found

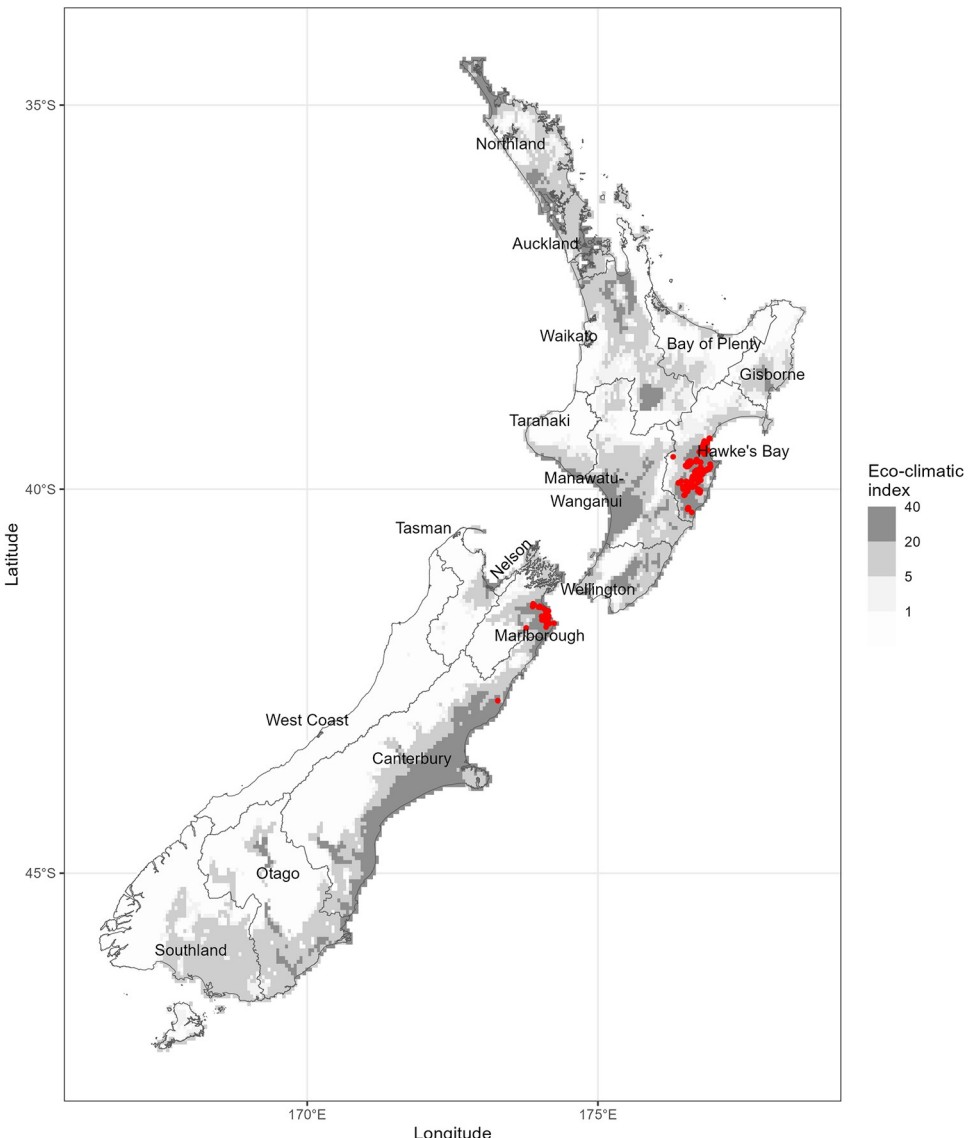

**Fig 1. Climate suitability of New Zealand for *N. neesiana*.** The map is derived from the published CLIMEX model for the species [16]. Major regional jurisdictions mentioned in the text are shown. Red symbols represent known occurrences. In the CLIMEX model, locations with an Eco-climatic index < 1.0 are climatically unsuitable while values > 1.0 represent locations that are increasingly more suitable. Reprinted from Bourdôt et al. (2010) [16] under a CC BY license, with permission from the New Zealand Plant Protection Society, original copyright 2010.

on a farm that received sheep from an infested farm 200 km away (Smith L, Environment Canterbury, *pers. comm.*) (Fig 1). Similarly, its introduction into the Hawkes Bay region in the North Island of New Zealand (Fig 1) has been attributed to seed for sowing sourced from an infested region [22].

In this contribution we show how region-based analyses of the 'cost of doing nothing' about *N. neesiana* underestimate the national benefit of preventing its spread in New Zealand. To that end, we extend nationally, a methodology developed for and applied to the Canterbury region [23] and consider the sensitivity of the estimated benefit to the discount rate used in its estimation.

## Method

To estimate the cost that would be incurred by the New Zealand pastoral sector should *N. nessiana*, spread throughout its potential range, we used the web app 'Cost Benefit Analysis for Regional Pest Management' [24]. The methodology deployed in the app is fully described elsewhere [23]. In essence, the method combines a logistic model of the weed's spread with empirically derived parameters (land area invaded currently, rate of spread, maximum land area susceptible to invasion) with estimates of the earnings ($/ha) from the susceptible land use type (pastoral farming), and reduction in earnings due to the weed, to estimate the present value (PV) of the pastoral sector productivity loss that would be avoided by preventing the weed's spread [23].

For each of New Zealand's 16 local government regions, we ran a 200-year simulation in which the weed was allowed to spread in the absence of any management that would otherwise prevent it from doing so. In these regional 'do-nothing' simulations, in which there are no management costs, the app returns the present value (PV) of the regional financial loss (cost) incurred over the simulation period due to the weed's spread within its climate niche in the region. It is calculated as the sum of the annual discounted costs that would be incurred under the 'do nothing' spread scenario. This 'cost' is the PV value of the lost pastoral production that would be avoided in the region by preventing the weed's spread. The loss is a result of grazing livestock being removed from infested paddocks during the summer when the *N. neesiana* is seeding [22]. This avoided loss equates to the PV 'benefit' of investing in regional programmes that prevent the weed's spread.

For each of the 16 simulations, region-specific settings were used for the input parameters '*Initial area infested*', '*Maximum area that could become infested*', and '*Earnings*' [24]. How these settings were estimated is discussed below. For the remaining parameters, the same setting was used for all regions: '*Time for infestation to reach 90% of maximum*' = 201 and 100 years; '*Reduction in earning caused by the pest*' = 25%; '*Discount rate*' = 3%. The rationale for these parameter settings is discussed in Bourdôt et al. [23]. The two spread rates are illustrated for the Hawkes Bay region (Fig 2); they encompass the range of values estimated from field records in the Marlborough region [23]. Here we used the moderately low discount rate of 3%, a rate considered appropriate for projects having significant intergenerational impact [25] and thus arguably appropriate for an invasive slow-spreading sleeper weed such as *N. neesiana*.

The region-specific settings for '*Initial area infested*' for two of the three regions with known infestations of *N. neesiana* (Hawke's Bay, Marlborough) were estimated using the current occurrences of the species in these regions and the method of buffered waypoints. By this method, the occurrences were grouped into polygons using a 5 m buffer and the hull algorithm with threshold = 0.003 decimal degrees. This results in occurrences that are close together being merged and outliers remaining as isolated circular polygons with a radius of 5 metres. The regional sums of all polygons and outliers probably give conservative estimates of the invaded land areas given that occurrence records provide an incomplete picture of the true extent of the weed. The method gave estimates of 2,557.8 ha infested in Marlborough (Jono Underwood, Marlborough District Council, *pers. comm.*) and 199.2 ha infested in Hawkes Bay (estimated based on data provided by Hawke's Bay Regional Council). For Canterbury, in the absence of a recently updated *N. neesiana* occurrence dataset for this region, we used an estimate of 330 ha currently infested (Laurence Smith, ECan, *pers. comm.*). For the remaining 13 regions in which *N. neesiana* is unknown (Fig 1), to enable an invasion trajectory to be calculated and 'do nothing' costs to be estimated, we assume that an unknown infestation of 1.0 ha exists.

The region-specific settings for '*Maximum area that could become infested*' were estimated as the area of currently climatically suitable High- and Low-producing land [26] within Land

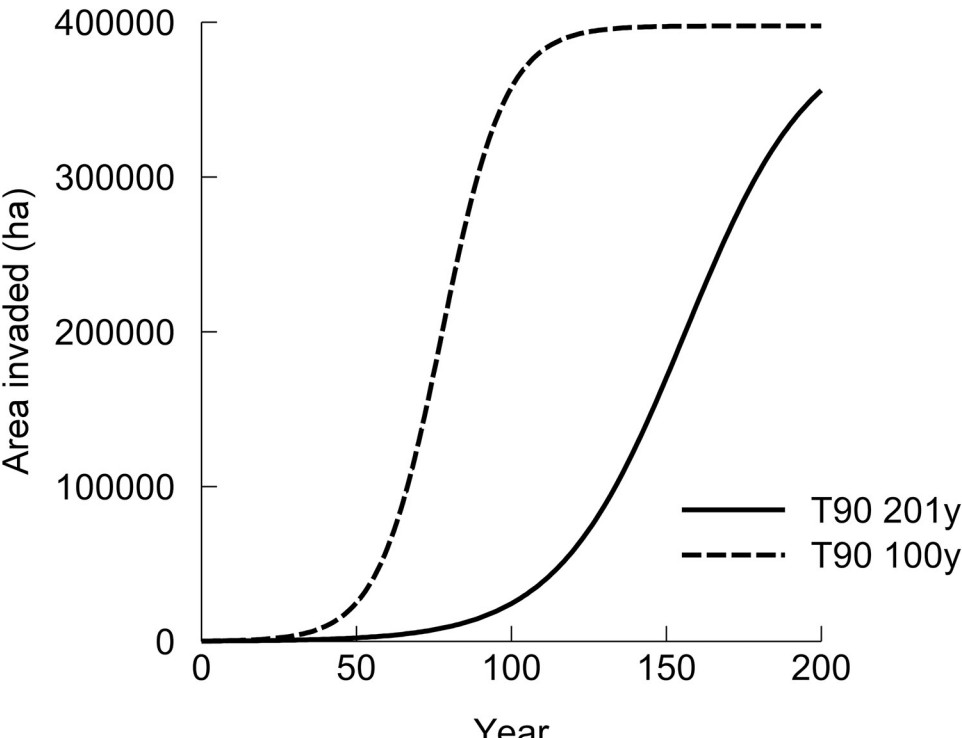

**Fig 2.** *N. neesiana* **invasion trajectories for the Hawke's Bay region assuming invasion rates where the time for the species to occupy 90% of its climatically suitable range (T90) are 201 or 100 years (y).** Initial area infested in the region, at Year = 0 (2021), is 199 ha and the maximum invadable area of climatically suitable pastureland is 397,578 ha. The trajectories were generated using the web application 'Cost Benefit Analysis for Regional Pest Management' [24].

Use Capability Classes 1 to 6 (land suitable for farming without physical limitations such as steep terrain and poor drainage) [27] that are in sheep, beef, or sheep/beef pasture as given for 2009 in the GIS agricultural database, AgriBase [28]. In determining these settings, we used the regional projections of the CLIMEX model for *N. neesiana* under current climate [16,18,23]. We note here that for regions where there have been increases or decreases since 2009 in land under sheep, beef or combined sheep and beef farming, our estimates of the benefits of stopping the spread of *N. neesiana* will be under- or over-estimates respectively. We also note that the estimated 3.96 million ha of susceptible pasture land nationally represents 26% of the total land area (15 million ha) that is climatically suitable for the species in New Zealand [16].

The region-specific settings for the parameter '*Earnings*' (Cash Operating Surplus in Bourdôt et al. [23]) were estimated as the weighted average of the 2021–22 profits ($/ha) forecasted for sheep, beef, and combined sheep and beef enterprises for Finishing/Breeding and Hill Country systems [29]. The weightings reflect the ratio of high to low producing pasture in the region and assume that 'Hill Country' systems are 'low-producing pasture' and 'Finishing/ Breeding' systems are 'high-producing pasture'.

In addition to estimating the cost of doing nothing about the spread of *N. neesiana*, we also estimated the breakeven cost of a Sustained Control management programme [30] for each of the 16 regions. Since this type of weed management programme aims to prevent spread beyond the currently infested areas [30], it can be considered a 'public good'. The management programmes' actual costs (not considered here) would include the cost of any research and/or technology development such as might be associated with determining appropriate regional management methods. The costs would also include all the management costs incurred by the

owners of land where the plant is established, ratepayers, and the regional authorities (councils) in promoting, implementing, assessing, and reporting on the programmes. The breakeven cost of a particular regional management programme is the cost that results in a Net Present Value (NPV) of zero in a cost benefit analysis [31]. We estimated this breakeven cost for each of the 16 regions using the 'Cost Benefit Analysis for Regional Pest Management' web app [24] by varying the level of management cost until the NPV was equal to zero.

To estimate the national cost of doing nothing about the spread of *N. neesiana*, and to carry out a sensitivity analysis for the discount rate, we set the '*Maximum area that could become infested*' to 3.96 million hectares, and tested three discount rates: 3%, 5% and 8%. These are commonly recommended discount rates for public projects where benefits accrue over longer, or shorter periods respectively [25]. The earnings per hectare were taken as the weighted average of the regional earnings per hectare (weighted as a proportion of the total area that could become infested).

## Results and discussion

The estimated PV cost of the *N. neesiana* invasions (the 'do nothing' cost) in each of the regions, and the respective regional breakeven management costs, are in Table 1. The 'do nothing' costs equate to the regional benefits of preventing the spread beyond the land areas that are currently infested (*Initial area infested* in Table 1). These estimates of the regional benefits of preventing the spread are likely to be conservative because: (1) the current areas occupied (i.e., *Initial area infested* in the CBA model) are underestimates due to the inevitable incompleteness of the occurrence data sets used to determine this parameter, and because (2) the estimates of *Maximum area that could become infested* are, for some regions, underestimates since they do not account for the estimated 60% increase in climatically suitable land by 2080 expected under climate change [18]. On the other hand, if the land area under pastoral farming declines in the future due to land-use changes, such as conversion to carbon farming forestry, then the 'do nothing' costs estimated here will be overestimates of the true regional costs to the pastoral sector of allowing *N. neesiana* to continue its spread.

The faster the assumed rate of invasion, the greater is the PV cost under a 'do nothing' scenario (Table 1). This effect occurs because the faster the invasion, the sooner the benefits of preventing it (avoided lost pastoral production costs) are realised and so the less is the effect of discounting. By contrast, the slower the invasion, the further out in time are the benefits of stopping it realised and thus the greater are the effects of discounting.

The 'do nothing' PV costs vary substantially between the regions driven largely by the differences in maximum areas invadable and differences in pastoral sector earnings per ha (Table 1). Looking at Hawke's Bay, for example, we see that the PV costs of *N. neesiana* spreading beyond its estimated 199 ha is NZ$16,861,417 million assuming an invasion rate of 201 years to invade 90% of the suitable land (Table 1). This rate of invasion is rather slow (Fig 2) but arguably realistic under current climate since it is the weed's invasion rate measured in the Marlborough region during the 18-year period from 1987 until 2005 [23,32]. Assuming the faster rate of invasion of 100 years to 90% occupation of invadable land (Fig 2), the PV cost of doing nothing increases to NZ$112,838,495 million (Table 1). It is conceivable this faster rate of invasion may be realised under future warmer climates due to more frequent floods and droughts which, respectively, would facilitate seed dispersal and establishment of new populations. So, the true cost of allowing the weed to spread in the region is likely to be somewhere between these two estimates (NZ$17–113 million).

Clearly, the three regions with known infestations of *N. neesiana* have a more urgent need to address the problem, regardless of any 'spillover' benefits that would accrue to regions from

**Table 1. Regional 'Do nothing' invasion costs (PV) and breakeven management costs for N. neesiana on climatically suitable sheep, beef, and sheep/beef farms in New Zealand (total of 3,960,549 ha), both calculated using the web app, 'Cost Benefit Analysis for Regional Pest Management'[23,24].**

| Region | Initial area infested (ha)[1] | Maximum area invadable (ha)[2] | Earnings ($/ha)[3] | Invasion cost PV ($)[4] $T_{90} = 201$ | Invasion cost PV ($)[4] $T_{90} = 100$ | Breakeven ($/year)[5] $T_{90} = 201$ | Breakeven ($/year)[5] $T_{90} = 100$ |
|---|---|---|---|---|---|---|---|
| *North Island* | | | | | | | |
| **Northland** | 1 | 222,636 | 658.47 | 7,537,165 | 104,866,453 | 226,400 | 3,154,205 |
| **Auckland** | 1 | 96,196 | 663.55 | 3,750,256 | 47,406,726 | 112,481 | 1,425,731 |
| **Bay of Plenty** | 1 | 20,124 | 663.12 | 1,054,653 | 10,801,095 | 31,394 | 324,581 |
| **Gisborne** | 1 | 92,453 | 290.68 | 1,589,552 | 19,997,366 | 47,671 | 601,404 |
| **Waikato** | 1 | 225,673 | 663.63 | 7,684,191 | 107,069,285 | 230,820 | 3,220,467 |
| **Manawatu-Wanganui** | 1 | 585,871 | 302.46 | 7,943,081 | 122,072,810 | 238,788 | 3,671,975 |
| **Hawke's Bay** | 199 | 397,578 | 290.68 | 16,861,417 | 112,838,495 | 478,293 | 3,365,422 |
| **Taranaki** | 1 | 27,133 | 302.46 | 609,366 | 6,521,606 | 18,179 | 196,028 |
| **Wellington** | 1 | 200,408 | 241.33 | 2,526,499 | 34,750,275 | 75,880 | 1,045,218 |
| *South Island* | | | | | | | |
| **Tasman** | 1 | 22,229 | 243.03 | 418,038 | 4,345,472 | 12,454 | 130,597 |
| **Nelson** | 1 | 925 | 243.51 | 43,198 | 240,368 | 1,178 | 7,109 |
| **Marlborough** | 2,558 | 136,012 | 207.62 | 22,831,542 | 53,521,246 | 421,260 | 1,344,450 |
| **Canterbury** | 330 | 790,257 | 209.88 | 22,825,150 | 158,972,670 | 651,983 | 4,747,497 |
| **Otago** | 1 | 644,929 | 158.73 | 4,530,480 | 70,278,045 | 136,204 | 2,113,986 |
| **Southland** | 1 | 498,125 | 158.73 | 3,622,651 | 54,793,422 | 108,895 | 1,648,186 |
| **West Coast**[6] | 0 | 0 | 0 | 0 | 0 | 0 | 0 |

[1]The area of sheep, beef, and sheep/beef pasture land (ha) currently invaded estimated as in Bourdôt et al., [23]. For each of the 13 regions currently without Chilean needle grass, an unknown infestation of 1.0 ha assumed.

[2]The maximum area of sheep, beef, and sheep/beef pastureland (ha) that is invadable as per Bourdôt et al., [23].

[3]The farm profit before tax (a weighted average for high- and low-producing pasture [26] using the 2021–22 forecasts for 'finishing/breeding' (high-producing land) and 'hill country' (low producing land) from the Sheep & Beef Farm Survey [29].

[4]Present value invasion cost estimated using the 'Do Nothing' scenario (setting *Yearly Costs* [of management] to zero) in the Cost Benefit Analysis for Regional Pest Management web application [24].

[5]Breakeven cost of preventing the spread of *N. neesiana* determined as the annual cost of management that returns a NPV = 0.0 for a Sustained Control programme using the Cost Benefit Analysis for Regional Pest Management web application [24].

[6]The West Coast region is climatically unsuitable for *N. neesiana* [16].

Scenarios assumed: Simulation period = 200 years; invasion rate $T_{90}$ = 201 or 100 years to 90% occupation of maximum invadable area; reduction in earnings due to the weed = 25%; discount rate = 3%. Dollar values are NZ$. Red font indicates regions with known infestations.

which the species is currently unknown (Table 1). From our calculations of the breakeven costs of regional management for *N. neesiana* using the slower rate of spread ($T_{90}$ = 201 years) (Table 1), the Hawkes Bay region could justify investing up to NZ$478,293 per year in spread prevention efforts. By contrast, Canterbury, with its much larger area of climatically suitable land and a similar size of the initial infested area, could justify the larger annual investment of NZ$651,983. Furthermore, if changing climate results in faster regional spread than the 201 years to occupy 90% of climatically suitable pastures, then considerably larger regional investments in managing the spread would be economically justifiable (Table 1).

With large areas of the country potentially vulnerable but currently without known populations of the weed, a national program could be justified. The annual breakeven cost of such a programme ranges from NZ$0.3 million to NZ$34.5 million depending on the rate of spread and discount rate (Table 2). In practice, it would be important to determine if the existing sites could be treated successfully with such investments; we briefly discuss this in our final

**Table 2. National 'Do nothing' invasion costs (PV) and breakeven management costs for N. neesiana on climatically suitable sheep, beef, and sheep/beef farms in New Zealand (total of 3,960,549 ha), both calculated using the web app, 'Cost Benefit Analysis for Regional Pest Management'[23,24].**

| Discount rate (%) | Initial area infested (ha)[1] | Maximum area invadable (ha)[2] | Earnings ($/ha)[3] | Invasion cost PV ($)[4] T90 = 201 | Invasion cost PV ($)[4] T90 = 100 | Breakeven ($/year)[5] T90 = 201 | Breakeven ($/year)[5] T90 = 100 |
|---|---|---|---|---|---|---|---|
| 3 | 3,099 | 3,960,549 | 285.55 | 192,377,089 | 1,160,129,685 | 5,344,521 | 34,455,915 |
| 5 | 3,099 | 3,960,549 | 285.55 | 32,464,015 | 226,965,034 | 1,180,835 | 10,906,448 |
| 8 | 3,099 | 3,960,549 | 285.55 | 9,780,740 | 39,232,505 | 340,000 | 2,696,141 |

[1]National total area of sheep, beef, and sheep/beef pastureland (ha) currently invaded (from Table 1).

[2]The maximum area of sheep, beef, and sheep/beef pastureland (ha) that is invadable (from Table 1).

[3]National average farm profit before tax (a weighted average for high- and low-producing pasture [26] using the 2021–22 forecasts for 'finishing/breeding' (high-producing land) and 'hill country' (low producing land) from the Sheep & Beef Farm Survey [29].

[4]Present value invasion cost estimated using the 'Do Nothing' scenario (setting *Yearly Costs* [of management] to zero) in the Cost Benefit Analysis for Regional Pest Management web application [24].

[5]Breakeven cost of preventing the spread of *N. neesiana* determined as the annual cost of management that returns a NPV = 0.0 for a Sustained Control programme using the Cost Benefit Analysis for Regional Pest Management web application [24].

Scenarios assumed: Simulation period = 200 years; invasion rate $T_{90}$ = 201 or 100 years to 90% occupation of maximum invadable area; reduction in earnings due to the weed = 25%; discount rate = 3%, 5% and 8%. Dollar values are NZ$.

paragraph. Taking the discount rate of 3% as being most appropriate, the annual breakeven cost for a national programme ranges from NZ$5.3 million to NZ$34.5 million. These results highlight that the devolution of pest management responsibilities to regional authorities leads to an under-estimation of the potential national losses from *N. neesiana* and thus also the benefits of a biosecurity response nationally. Importantly any national or regional program would be expected to run for many years, and a dedicated team of committed successive professionals would need to keep the prevention benefits in mind over decades [5]. It would require a strong commitment to early detection, land-owner extension, record-keeping, and follow-up for infested sites. Such long-term and community-minded thinking is hard to maintain in political institutions, such as regional councils, that need to justify this ongoing expenditure to ratepayers given multiple competing needs. In the case of *N. neesiana*, the animal-welfare related impacts of this weed that occur when farmers fail to remove livestock from infested pastures during the summer months when the sharp seeds are being shed, provide an added incentive for management.

Three key assumptions are fundamental to our model [23]. First, that the invasion of New Zealand by *N. nessiana* would continue unimpeded in the absence of regionally co-ordinated landscape-scale management programmes. Second, that to prevent livestock-mediated seed dispersal, and the animal welfare and product quality impacts of the seeds, invaded paddocks must be rested from grazing during the 3-month seeding period, resulting in a 25% p.a. loss in revenue. Third, that regionally coordinated management can prevent the spread. This assumption is supported by experiments in New Zealand showing low pasture reinfestation potentials from soil-borne seeds within a few years of repeated herbicide and soil cultivation treatments [15], and a propensity for slow spread in the absence of management [23]. The latter can be attributed to the predominantly short-distance dispersal of the seeds. The seeds clump together on the panicle at maturity due to the intertwining of their hygroscopic awns and then fall to the ground at the base of the parent plant. Their long-distance dispersal is likely to occur only through human activities including the movement of seed-contaminated farm animals, forages, hay, soil, and river gravel, and of farm and roadside maintenance machinery [14]. In Australia, local eradication of *N. neesiana* has proven possible providing seeding is prevented

using regular applications of herbicides such as flupronate and glyphosate for 7–10 years (*pers. comm*. Steve Taylor, ACT Parks & Conservation Service, Australia). This confirms the predictions of the soil seed decay model developed in New Zealand [15]. Altogether, these ecological traits and management vulnerabilities indicate that preventing the geographic spread of *N. neesiana* in New Zealand is feasible. Regionally effective containment would depend on detection and sustained control efforts over many years [33]. The costs of these control methods would need to be estimated, shown to be equal to or less than the breakeven costs we have identified, and affordable.

## Acknowledgments

We thank biosecurity staff at Hawke's Bay Regional Council for stimulating our interest in the question addressed in the paper and Graeme Doole at AgResearch for helpful comments on the manuscript.

## Author Contributions

**Conceptualization:** Graeme W. Bourdôt, Christopher E. Buddenhagen.

**Formal analysis:** Graeme W. Bourdôt.

**Funding acquisition:** Graeme W. Bourdôt, Christopher E. Buddenhagen.

**Methodology:** Graeme W. Bourdôt, Christopher E. Buddenhagen.

**Writing – original draft:** Graeme W. Bourdôt, Christopher E. Buddenhagen.

**Writing – review & editing:** Graeme W. Bourdôt, Christopher E. Buddenhagen.

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
