## [Decision Letter · Decision Letter 0]

5 Sep 2023

PONE-D-23-24294The cost of doing nothing about a sleeper weed –Nassella neesiana in New ZealandPLOS ONE

Dear Dr. Bourdôt,

Thank you for submitting your manuscript to PLOS ONE. After careful consideration, we feel that it has merit but does not fully meet PLOS ONE’s publication criteria as it currently stands. Therefore, we invite you to submit a revised version of the manuscript that addresses each and every point raised during the review process.

We look forward to receiving your revised manuscript.

Kind regards,

Long Chu

Academic Editor

PLOS ONE

Journal Requirements:

3. We note that Figures 1 and 3 in your submission contain map/satellite images which may be copyrighted. All PLOS content is published under the Creative Commons Attribution License (CC BY 4.0), which means that the manuscript, images, and Supporting Information files will be freely available online, and any third party is permitted to access, download, copy, distribute, and use these materials in any way, even commercially, with proper attribution. For these reasons, we cannot publish previously copyrighted maps or satellite images created using proprietary data, such as Google software (Google Maps, Street View, and Earth). For more information, see our copyright guidelines: http://journals.plos.org/plosone/s/licenses-and-copyright.

1.) You may seek permission from the original copyright holder of Figures 1 and 3 to publish the content specifically under the CC BY 4.0 license.  

2.) If you are unable to obtain permission from the original copyright holder to publish these figures under the CC BY 4.0 license or if the copyright holder’s requirements are incompatible with the CC BY 4.0 license, please either i) remove the figure or ii) supply a replacement figure that complies with the CC BY 4.0 license. Please check copyright information on all replacement figures and update the figure caption with source information. If applicable, please specify in the figure caption text when a figure is similar but not identical to the original image and is therefore for illustrative purposes only.

Additional Editor Comments:

Both reviewers consider the analysis competent. However, Reviewer 1 is critical of the lack of empirical or conceptual innovation which, after reading the manuscript multiple times, I presume you are able to address or clarify. In doing so, it is important to demonstrate how your work is connected to recent studies on the economics of controlling invasive weeds or sleeper weeds.

There are additional comments and suggestions from the reviewers that will contribute to the overall improvement of the paper.

Reviewers' comments:

Reviewer's Responses to Questions

**Comments to the Author**

1. Is the manuscript technically sound, and do the data support the conclusions?

Reviewer #1: Partly

Reviewer #2: Yes

2. Has the statistical analysis been performed appropriately and rigorously? 

Reviewer #1: N/A

Reviewer #2: Yes

3. Have the authors made all data underlying the findings in their manuscript fully available?

Reviewer #1: Yes

Reviewer #2: Yes

4. Is the manuscript presented in an intelligible fashion and written in standard English?

Reviewer #1: Yes

Reviewer #2: Yes

5. Review Comments to the Author

Reviewer #1: The authors develop a spread model for a pasture weed, Chilean needle grass (Nassella neesiana), to estimate the costs and benefits of activities aimed at slowing the weed’s spread. The authors state that a containment approach is suitable (page 10), and estimate the investment in management that would be justified to avoid the costs of uncontrolled spread.

Although the analysis is competently done, the paper lacks significant empirical or conceptual innovation. It relies on assumed spread rates rather than empirically estimated rates its assumptions about the capacity to slow spread are not clearly stated or adequately supported with evidence. There also is a lack of information on the capacity of landowners to mitigate costs arising from the invasive weed by modifying production practices or using chemical or other control methods – this is needed to provide a better understanding of the costs of uncontrolled spread. The report’s main finding has been made elsewhere, namely, that considering part of the potential geographic range of a non-indigenous species will underestimate the benefits of slowing spread or eradicating the species.

It may be possible to extend the analysis to increase the generality and practical utility of its findings. One possible approach would be to consider a broader range of hypothetical spread rates within a credible range, and estimate a relationship between the benefits of containment and the assumed rate of spread under specific assumptions on the extent to which spread can be slowed with containment. This would require further thought and additional simulations. I suggest this as one of many possible ways to enhance the conceptual or empirical innovation of the paper commensurate with the level of innovation required of a journal such as Plos One.

Reviewer #2: General comments:

This manuscript statistically assesses the present value of future invasion costs of Nassella neesiana under a 'doing nothing' scenario, as well as the associated breakeven costs, across 16 local regions in New Zealand. It also accounts for the impacts of climate change by changing invasion rate. The paper underscores the cost-effectiveness of a nationwide approach to managing this pest. The objectives are well-defined, and the study employs multiple evaluations to substantiate its conclusions. The results are presented with scientific rigor.

Specific comments:

1) Line 27: Change ‘regional’ to ‘local’, as mentioned in line 23 ‘sixteen local government regions’.

2) Line 62: Given the manuscript's focus on New Zealand, consider whether it's necessary to mention Australia here and in line 67, as well as lines 78-82.

3) Line 83: Consider briefly explaining the terminology 'Total Control,' 'Containment,' and 'Sustained Control' to clarify the status of the pest in the infested area.

4) Line 91: Does 'they' refer to wild animals?

5) Line 100: ‘we extend a methodology developed for…’ What is this methodology about? Could you provide more information about the methodology mentioned here?

6) Line 107: ‘… is fully described elsewhere [19].’ Consider providing a brief description of the methodology instead of directing readers to search for it in the literature.

7) Line 110, 111,115: Change ‘regional’ to ‘local’.

8) Line 111: Remove the '$' symbol.

9) Line 115: Move the sentence ‘It is calculated as the sum of …’ to line 113, after ‘… niche in the region.’

10) Line 119 to line 127: Consider moving the sentences 'For the remaining parameters…' to a new paragraph before line 167.

11) Line 137: Change ‘was’ to ‘were’.

12) Line 141: Change ‘estimated by Chris Buddenhagen from data provided by…’ to ‘estimated based on data provided by…’

13) Line 150 &151: It appears that the authors have included a 5m buffer area twice. Please clarify.

14) Line 155-159: this sentence is too long. Consider breaking it into shorter, more concise sentences.

15) Line 155: Move ‘(3.96 million ha nationally)’ to before ‘We also note that the 3.96 million ha of…’ in line 164.

16) Line 161: ‘The methodology is given in full in Bourdôt et al [19].’ Again, consider providing a brief description of the methodology mentioned here or simply include it as a reference.

17) Line 162: ‘We note here that for regions where there have been increases or decreases since 2009…’ If this statement reflects your findings, consider moving it to the Results and Discussion section.

18) Line 180 &181: Change ‘regional’ to ‘local’.

19) Table 1: In the figure caption, provide an explanation for the special values assigned to the ‘West Coast’.

20) Line 237: Change ‘regional’ to ‘local’.

21) Line 238-239: Change ‘while Canterbury, with its much larger area of climatically suitable land…’ to ‘while Canterbury, with its much larger area of climatically suitable land and a similar size of the initial infested area…’

22) Line 241: ‘$22,825,150’. The figure should be '$651,983' based on Table 1.

23) Line 243: ‘Furthermore, if we believe that changing climate will result in faster regional spread than the 201 years to occupy 90% of climatically suitable pastures measured in Marlborough, then…’. This is also the case in the other two regions.

24) Line 247: Remove ‘/year’.

25) Line 251 &258: Change ‘regional’ to ‘local’.

6. PLOS authors have the option to publish the peer review history of their article (what does this mean?). If published, this will include your full peer review and any attached files.

Reviewer #1: No

Reviewer #2: No

---

## [Author Response · Author response to Decision Letter 0]

11 Oct 2023

1. Please upload the completed Content Permission Form or other proof of granted permissions as an ""Other"" file with your submission. <Done.>

In the figure caption of the copyrighted figure, please include the following text: “Reprinted from [ref] under a CC BY license, with permission from [name of publisher], original copyright [original copyright year].” <Done.>

2. Both reviewers consider the analysis competent. However, Reviewer 1 is critical of the lack of empirical or conceptual innovation which, after reading the manuscript multiple times, I presume you are able to address or clarify. < We think the analytical approach we use provides insights into the level of investment that could be justified for a sleeper pest containment or eradication programme. We also quantify the extent to which the benefits of such measures can be underestimated where pest management is devolved to areas or jurisdictions smaller than the range where the impacts are realized.> In doing so, it is important to demonstrate how your work is connected to recent studies on the economics of controlling invasive weeds or sleeper weeds. <We highlight some existing work where weed management of incipient invaders is framed in terms of optimal acceptable density that minimizes impact costs relative to management costs. This contrasts with our approach where containment and eradication programmes are valued more as a prevention measure. See our next text on lines 61-68 in the clean version of our revision. Some economics analyses have shown that prevention is the most cost-effective strategy to manage invasive species. >

3. 1. Is the manuscript technically sound, and do the data support the conclusions?

Reviewer #1: Partly <See question 4.>

4. Reviewer #1: The authors develop a spread model for a pasture weed, Chilean needle grass (Nassella neesiana), to estimate the costs and benefits of activities aimed at slowing the weed’s spread. The authors state that a containment approach is suitable (page 10), and estimate the investment in management that would be justified to avoid the costs of uncontrolled spread. <Although we do develop a spread model for Chilean needle grass, we have not used it here to estimate the costs and benefits of activities aimed at slowing the weed’s spread. Rather, we have used it to estimate the ‘cost of doing nothing’ about managing spread (i.e., allowing the spread to occur). This cost is equivalent to the benefit of stopping the spread. This estimate of the potential damages is required for any cost-benefit analysis. We do not identify all the specific costs of management because they vary across the weed’s range. We recognise that slowing a weed’s spread, rather than stopping it, is a common outcome of under-resourced management. But we doubt that slowing spread would ever be an acceptable goal of a publicly funded weed control programme. Our aim was to estimate the Present Value benefits of preventing Chilean needle grass from spreading within and between regions in New Zealand and to estimate the investment that could be justified to achieve that. It was also our aim to illustrate in a quantitative manner, the benefits of national as compared to regional decision-making regarding the management of this invasive weed. Please see our new text on lines 134-139 and 145-151.>

Although the analysis is competently done, the paper lacks significant empirical or conceptual innovation. It relies on assumed spread rates rather than empirically estimated rates <(our rate of T90=210 years is an empirical estimate sourced from the article [22] whilst the faster rate of T90=100 is included to illustrate the pronounced effect that invasion speed has in a discounted analysis of costs and the potential effects of a warming climate on the weed’s spread in the absence of management)> its assumptions about the capacity to slow spread are not clearly stated or adequately supported with evidence <(we make no such assumptions)>. There also is a lack of information on the capacity of landowners to mitigate costs arising from the invasive weed by modifying production practices or using chemical or other control methods – this is needed to provide a better understanding of the costs of uncontrolled spread <We have added some text at lines 340-348 to show that there are tools (mowing and glyphosate) that should lead to local extirpation of populations over 7-10 years. Weed impact mitigation costs (control, or practice change) would vary with density and stage of invasion. Even if they could be estimated, they would add to the cost of control. But we are not considering this weed control cost in our analysis.> The report’s main finding has been made elsewhere, namely, that considering part of the potential geographic range of a non-indigenous species will underestimate the benefits of slowing spread or eradicating the species <This does seem self-evident, but our work comments about the role of policy and management (devolved responsibility) and quantifies the problem. The reviewer did not point toward an article that is similar to ours – we have found and cited other economic analyses. Perhaps they are referring here to our paper “Bourdôt GW, Basse B, Kriticos DJ, Dodd M. A cost-benefit analysis blueprint for regional weed management: Nassella neesiana (Chilean needle grass) as a case study. New Zealand Journal of Agricultural Research. 2015;58(3):1-14. doi: 10.1080/00288233.2015.1037460.” In that paper we did comment that our analysis of the costs and benefits of regional management of the species in the Canterbury region underestimates the national benefit. In this new analysis we are quantifying the monetary value of the cost of doing nothing at the national scale and comparing that to the separate regional costs of doing nothing to show the extent to which regional analyses underestimate the national cost and hence the benefit of preventing spread.>

It may be possible to extend the analysis to increase the generality and practical utility of its findings. One possible approach would be to consider a broader range of hypothetical spread rates within a credible range and estimate a relationship between the benefits of containment and the assumed rate of spread under specific assumptions on the extent to which spread can be slowed with containment. <We have previously published such an analysis of the model in the Bourdôt et al (2015) paper mentioned above (Figure 3 copied below), where its sensitivity to the spread rate in the absence of control and other model parameters is revealed, but with the assumption that spread is prevented. Regional and national management programmes that simply slow the spread of a weed are not acceptable under the NZ Biosecurity Act even though in practice that may be the outcome. This act provides for only four categories of pest management at regional and national scales all aimed at preventing spread: Exclusion, Eradication, Progressive Containment, Sustained Control.> This would require further thought and additional simulations. <Yes, it would, but is beyond the scope of our project and would be of theoretical value only.> I suggest this as one of many possible ways to enhance the conceptual or empirical innovation of the paper commensurate with the level of innovation required of a journal such as Plos One. <Our emphasis is on the implications of jurisdictional and cost-benefit approaches to biosecurity policy setting for sleeper weeds. We are highlighting the following:

1) Sleeper pest management is more appropriately regarded as a means of impact prevention than impact mitigation as required under the NZ Biosecurity Act.

2) Devolved invasive management to sub-regions likely (inevitably) leads to under-investment in the management of sleeper pests and weeds.

3) A national or multi-region approach to the cost-benefit assessment is more appropriate for sleeper weeds, even for those that would slowly spread in the absence of management and if their current range is regionally restricted.

Based on this reviewer’s suggestions, we ran some simulations to consider a broader range of hypothetical spread rates and the benefits of containment (graphs below). We find this analysis provides little insight into the question we are addressing.

For the Canterbury region, the relationship between the net benefit of containment of the weed and the initial size of the infestation (A0) and the Invasion speed (A), Impact on pastoral production (B) and discount rate (C) has been explored (graph below from Bourdôt et al (2015))

Our current analysis offers new insights into the problems of a regional approach, extends the method to all regions, and reveals the expenditure that could be justified in preventing spread nationally.>

5. Reviewer #2: General comments:

This manuscript statistically assesses the present value of future invasion costs of Nassella neesiana under a 'doing nothing' scenario, as well as the associated breakeven costs, across 16 local regions in New Zealand. It also accounts for the impacts of climate change by changing invasion rate. The paper underscores the cost-effectiveness of a nationwide approach to managing this pest. The objectives are well-defined, and the study employs multiple evaluations to substantiate its conclusions. The results are presented with scientific rigor.

<We thank this reviewer for their positive comments.>

Specific comments:

1) Line 27: Change ‘regional’ to ‘local’, as mentioned in line 23 ‘sixteen local government regions’. <We have inserted ‘local’ and placed ‘regional’ in brackets to clarify that local government occurs in regions – this is a specific type of local government in NZ. We use ‘region’ or ‘regional’ from here on rather than local. In New Zealand, we have Regional and District councils, which are two types of local government.>

2) Line 62: Given the manuscript's focus on New Zealand, consider whether it's necessary to mention Australia here and in line 67, as well as lines 78-82. <Reference to Australia removed.>

3) Line 83: Consider briefly explaining the terminology 'Total Control,' 'Containment,' and 'Sustained Control' to clarify the status of the pest in the infested area. <We believe that our original text on lines 85-86 “these classifications requiring landholders to eradicate, contain and control the species respectively.” provides this clarification.>

4) Line 91: Does 'they' refer to wild animals? <We have revised this to clarify that it is the fruit that are dispersed in water. >

5) Line 100: ‘we extend a methodology developed for…’ What is this methodology about? Could you provide more information about the methodology mentioned here? <Have provided some basic details of the method in the first paragraph of the Methods section.>

6. 6) Line 107: ‘… is fully described elsewhere [19].’ Consider providing a brief description of the methodology instead of directing readers to search for it in the literature. <Have provided some basic details of the method.>

7) Line 110, 111,115: Change ‘regional’ to ‘local’. <We prefer to use the term ‘regional’ since that is the scale (millions of hectares) at which pest management plans are made in New Zealand. Local implies a much smaller scale (e.g., individual catchment of farm) at which our analyses do not apply.>

8) Line 111: Remove the '$' symbol. <Done.>

9) Line 115: Move the sentence ‘It is calculated as the sum of …’ to line 113, after ‘… niche in the region.’ <Done.>

10) Line 119 to line 127: Consider moving the sentences 'For the remaining parameters…' to a new paragraph before line 167. <We considered making that shift but believe our meaning is clearer with the original sentence layout explaining the settings common to all simulation first.>

11) Line 137: Change ‘was’ to ‘were’. <Corrected.>

12) Line 141: Change ‘estimated by Chris Buddenhagen from data provided by…’ to ‘estimated based on data provided by…’ <Done.>

13) Line 150 &151: It appears that the authors have included a 5m buffer area twice. Please clarify.

14) Line 155-159: this sentence is too long. Consider breaking it into shorter, more concise sentences. <We have removed this Figure 3 from the paper, considering it unnecessary. This also removed the copyright issues with the aerial imagery.>

15) Line 155: Move ‘(3.96 million ha nationally)’ to before ‘We also note that the 3.96 million ha of…’ in line 164. <Done.>

16) Line 161: ‘The methodology is given in full in Bourdôt et al [19].’ Again, consider providing a brief description of the methodology mentioned here or simply include it as a reference. <We have removed this text and added the reference to the previous sentence.>

17) Line 162: ‘We note here that for regions where there have been increases or decreases since 2009…’ If this statement reflects your findings, consider moving it to the Results and Discussion section. <We have left this text here in the Methods section since it is not a result but rather a caveat regarding the results.> 

18) Line 180 &181: Change ‘regional’ to ‘local’. <We have not done this for the reason given above at point 7.>

19) Table 1: In the figure caption, provide an explanation for the special values assigned to the ‘West Coast’. <Done.>

20) Line 237: Change ‘regional’ to ‘local’. <We have not done this for the reason given above at point 7.>

21) Line 238-239: Change ‘while Canterbury, with its much larger area of climatically suitable land…’ to ‘while Canterbury, with its much larger area of climatically suitable land and a similar size of the initial infested area…’ <Alteration made.>

22) Line 241: ‘$22,825,150’. The figure should be '$651,983' based on Table 1. <We have made that correction and thank the reviewer very much for spotting our mistake.>

23) Line 243: ‘Furthermore, if we believe that changing climate will result in faster regional spread than the 201 years to occupy 90% of climatically suitable pastures measured in Marlborough, then…’. This is also the case in the other two regions. <Correct. We have modified the text accordingly.>

24) Line 247: Remove ‘/year’. <Done.>

25) Line 251 &258: Change ‘regional’ to ‘local’. <We have not done this for the reason given above at point 7.>

---

## [Decision Letter · Decision Letter 1]

16 Nov 2023

PONE-D-23-24294R1The cost of doing nothing about a sleeper weed –Nassella neesiana in New ZealandPLOS ONE

Dear Dr. Bourdôt,

Thank you for submitting your manuscript to PLOS ONE. After careful consideration, we feel that it has merit but does not fully meet PLOS ONE’s publication criteria as it currently stands. Therefore, we invite you to submit a revised version of the manuscript that addresses the points raised during the review process.

We look forward to receiving your revised manuscript.

Kind regards,

Long Chu

Academic Editor

PLOS ONE

Journal Requirements:

Additional Editor Comments:

Your article has a very good chance of being accepted for publication, but a reviewer has requested minor amendments to be made. Please respond to the reviewer’s request.

Reviewers' comments:

Reviewer's Responses to Questions

**Comments to the Author**

1. If the authors have adequately addressed your comments raised in a previous round of review and you feel that this manuscript is now acceptable for publication, you may indicate that here to bypass the “Comments to the Author” section, enter your conflict of interest statement in the “Confidential to Editor” section, and submit your "Accept" recommendation.

Reviewer #1: (No Response)

2. Is the manuscript technically sound, and do the data support the conclusions?

Reviewer #1: Yes

3. Has the statistical analysis been performed appropriately and rigorously? 

Reviewer #1: Yes

4. Have the authors made all data underlying the findings in their manuscript fully available?

Reviewer #1: Yes

5. Is the manuscript presented in an intelligible fashion and written in standard English?

Reviewer #1: Yes

6. Review Comments to the Author

Reviewer #1: The paper highlights the potential for underinvestment in containment programs in countries with regional jurisdictions. In particular, if spread to another jurisdiction is ignored, there may be underinvestment. This relies on the untested assumption that spending more on containment could prevent geographic spread. Since this is probably a rare circumstance, the reliance on a strong assumption weakens the practical significance of the paper. Nonetheless, the issue of underinvestment in containment of invasive species in countries with regional governments has received little attention and should be raised. This is the main strong point of the paper. Its weak point is the lack of empirical support that the case study has broader significance. A second concern I have is that if the pest causes complete loss of access to infested paddocks, and if early paddock detection can prevent this, the model would have overstated the costs of allowing the species to spread. It is implicit in the paper that the species can be contained, which makes it important to address this concern in the revised manuscript. Subject to explicitly stating the strong assumption made about the capacity to prevent spread, and the cost-estimation assumption that infestation would go u8nimpeded until paddock access is lost, I recommend that the paper be accepted.

7. PLOS authors have the option to publish the peer review history of their article (what does this mean?). If published, this will include your full peer review and any attached files.

Reviewer #1: No

---

## [Author Response · Author response to Decision Letter 1]

23 Nov 2023

Reviewer #1: The paper highlights the potential for underinvestment in containment programs in countries with regional jurisdictions. In particular, if spread to another jurisdiction is ignored, there may be underinvestment. This relies on the untested assumption that spending more on containment could prevent geographic spread. Since this is probably a rare circumstance, the reliance on a strong assumption weakens the practical significance of the paper. Nonetheless, the issue of underinvestment in containment of invasive species in countries with regional governments has received little attention and should be raised. This is the main strong point of the paper. Its weak point is the lack of empirical support that the case study has broader significance. A second concern I have is that if the pest causes complete loss of access to infested paddocks, and if early paddock detection can prevent this, the model would have overstated the costs of allowing the species to spread. It is implicit in the paper that the species can be contained, which makes it important to address this concern in the revised manuscript. Subject to explicitly stating the strong assumption made about the capacity to prevent spread, and the cost-estimation assumption that infestation would go u8nimpeded until paddock access is lost, I recommend that the paper be accepted.

<We thank the reviewer for these helpful comments. We have addressed them in the new paragraph on Lines 295-314 in the track changes version of our resubmitted manuscript – copied below – and have made relevant changes to the final paragraph on Lines 315-323.>

<” Three key assumptions are fundamental to our model [23]. First, that the invasion of New Zealand by N. nessiana would continue unimpeded in the absence of regionally co-ordinated landscape-scale management programmes. Second, that to prevent livestock-mediated seed dispersal, and the animal welfare and product quality impacts of the seeds, invaded paddocks must be rested from grazing during the 3-month seeding period, resulting in a 25% p.a. loss in revenue. Third, that regionally coordinated management can prevent the spread. This assumption is supported by experiments in New Zealand showing low pasture reinfestation potentials from soil-borne seeds within a few years of repeated herbicide and soil cultivation treatments [15], and a propensity for slow spread in the absence of management [23]. The latter can be attributed to the predominantly short-distance dispersal of the seeds. The seeds clump together on the panicle at maturity due the intertwining of their hygroscopic awns and then fall to the ground at the base of the parent plant. Their long-distance dispersal is likely to occur only through human activities including the movement of seed-contaminated farm animals, forages, hay, soil, and river gravel, and of farm and roadside maintenance machinery [14]. In Australia, local eradication of N. neesiana has proven possible providing seeding is prevented using regular applications of herbicides such as fluproponate and glyphosate for 7-10 years (pers. comm. Steve Taylor, ACT Parks & Conservation Service, Australia). This confirms the predictions of the soil seed decay model developed in New Zealand [15]. Altogether, these ecological traits and management vulnerabilities indicate that preventing the geographic spread of N. neesiana in New Zealand is feasible.Regionally effective containment would depend on detection and sustained control efforts over many years [33]. The costs of these control methods would need to be estimated, shown to be equal to or less than the breakeven costs we have identified, and affordable.”>

---

## [Editor Report · Decision Letter 2]

27 Nov 2023

The cost of doing nothing about a sleeper weed –Nassella neesiana in New Zealand

PONE-D-23-24294R2

Dear Dr. Bourdôt,

We’re pleased to inform you that your manuscript has been judged scientifically suitable for publication and will be formally accepted for publication once it meets all outstanding technical requirements.

Kind regards,

Long Chu

Academic Editor

PLOS ONE
---

## [Editor Report · Acceptance letter]

4 Dec 2023

PONE-D-23-24294R2 

The cost of doing nothing about a sleeper weed –*Nassella neesiana* in New Zealand 

Dear Dr. Bourdôt:

I'm pleased to inform you that your manuscript has been deemed suitable for publication in PLOS ONE. Congratulations! Your manuscript is now with our production department. 

Kind regards, 

on behalf of

Dr. Long Chu 

Academic Editor

PLOS ONE